# Construction of a synthetic metabolic pathway for biosynthesis of the non-natural methionine precursor 2,4-dihydroxybutyric acid

Thomas Walther[1,2,†], Christopher M. Topham[1], Romain Irague[1], Clément Auriol[1,2], Audrey Baylac[1], Hélène Cordier[2], Clémentine Dressaire[1,2], Luce Lozano-Huguet[1], Nathalie Tarrat[1,†], Nelly Martineau[1,3], Marion Stodel[1], Yannick Malbert[1,2], Marc Maestracci[3], Robert Huet[3], Isabelle André[1,2], Magali Remaud-Siméon[1,2] & Jean Marie François[1,2]

2,4-Dihydroxybutyric acid (DHB) is a molecule with considerable potential as a versatile chemical synthon. Notably, it may serve as a precursor for chemical synthesis of the methionine analogue 2-hydroxy-4-(methylthio)butyrate, thus, targeting a considerable market in animal nutrition. However, no natural metabolic pathway exists for the biosynthesis of DHB. Here we have therefore conceived a three-step metabolic pathway for the synthesis of DHB starting from the natural metabolite malate. The pathway employs previously unreported malate kinase, malate semialdehyde dehydrogenase and malate semialdehyde reductase activities. The kinase and semialdehyde dehydrogenase activities were obtained by rational design based on structural and mechanistic knowledge of candidate enzymes acting on sterically cognate substrates. Malate semialdehyde reductase activity was identified from an initial screening of several natural enzymes, and was further improved by rational design. The pathway was expressed in a minimally engineered Escherichia coli strain and produces 1.8 g l$^{-1}$ DHB with a molar yield of 0.15.

[1] LISBP, Université de Toulouse, CNRS, INRA, INSA, Toulouse, France, 135 Avenue de Rangueil, F-31077 Toulouse, France. [2] TWB, 3 rue des Satellites, Canal Biotech Building 2, Toulouse F-31400, France. [3] Adisseo SA, Antony Parc II, 10 Place General de Gaulle, Antony F-92160, France. † Present address(es): Institute of Natural Materials Technology, TU Dresden, 01062 Dresden, Germany (T.W.); CEMES-CNRS, 29 Rue Jeanne Marvig, F-31055 Toulouse, France (N.T.). Correspondence and requests for materials should be addressed to J.M.F. (email: fran_jm@insa-toulouse.fr).

The growing shortage of fossil raw materials has prompted an increasing economic and ecological interest in replacing petrol-based chemical syntheses by biochemical processes that rely on the utilization of renewable resources[1–3]. The amino-acid product family provides a particularly striking example of our growing technological ability to replace petrol-based syntheses of bulk chemicals by microbial production processes. Most of the 20 proteinogenic amino acids are currently synthesized by fermentation processes using renewable glucose as the primary carbon source[4]. A notable exception is the essential amino-acid methionine and its analogue 2-hydroxy-4-(methylthio)butyrate (HMTB), which are still exclusively produced from petrol[5]. The cumulated annual volume of methionine and HMTB production reached ∼1 million tons in 2014, the largest part of which is used as a supplement in animal diet, greatly increasing the nutritional value of feed stocks[6–8]. Methionine production is thus an important cornerstone in the satisfaction of the ever growing demand of the human population for animal protein[9]. The development of sustainable means of methionine or HMTB production from renewable resources is therefore of significant importance.

Economically viable methionine and HMTB fermentation processes are currently lacking, which is mainly due to the extremely high metabolic cost of incorporating sulfur into these molecules[10]. A promising alternative to the microbial production of HMTB is to envisage a two-stage process whereby the fermentative production of a functionalized carbohydrate precursor is followed by the chemical incorporation of sulfur. For example, by employing established chemistry, it is possible to convert 2,4-dihydroxybutyric acid (DHB) and methanethiol into HMTB with 100% carbon yield[11]. However, while the chemical synthesis of DHB from petrol-derived γ-butyrolactone is feasible[11], it is not economically viable.

Although trace amounts of DHB have been identified in human patients suffering from succinic semialdehyde dehydrogenase deficiency[12], no natural metabolic pathway to access DHB has ever been described. Thus, DHB is a metabolite that cannot be efficiently synthesized via currently annotated natural metabolic networks. These considerations have motivated our exploration of plausible non-natural pathways leading to DHB that take advantage of synthetic biology approaches. We herein present a new synthetic DHB pathway whose chemical logic was inspired by the biosynthesis of homoserine. This amino acid is naturally produced by the successive action of aspartate kinase (AK), aspartate-β-semialdehyde dehydrogenase (ASD), and homoserine dehydrogenase (HSD). Homoserine and DHB are structurally very similar molecules and differ only in the presence of an amino or hydroxyl group, respectively, on the carbon 2 position. Since the amino groups of the homoserine pathway intermediates are not involved in the catalytic reaction mechanism of the individual enzymes, we considered it feasible to apply the (bio)chemical principle of the natural homoserine pathway to convert malate into DHB. However, the existence of neither the required malate kinase (MK), malate-β-semialdehyde dehydrogenase (MSD), and malate semialdehyde reductase (MSR) enzymatic activities, nor the malyl-4-phosphate (malyl-P) and malate-4-semialdehyde pathway intermediates has been previously reported[13]. Thus, it was not possible to use established approaches in which the synthetic pathway is assembled by harvesting genes encoding the required activities from different biological sources and expressing them together in a single production organism[14–16].

We therefore set out to create the required enzymatic activities by computer-aided engineering of template enzymes acting on sterically cognate substrates but which displayed little or no activity towards the synthetic pathway intermediates. We used structural analysis and molecular modelling methods to identify target residue sites for mutation and designed mutant libraries of moderate size that were screened for the isolation of mutants with the requisite enzymatic properties. We have obtained all three enzymatic activities and demonstrated the production of $1.8 \, g \, l^{-1}$ DHB from an initial glucose concentration of $20 \, g \, l^{-1}$ by expressing the synthetic pathway in an E. coli strain.

## Results

**Thermodynamic feasibility and maximum DHB yield.** The proposed pathway proceeds through the activation of the malate β-carboxylate group by phosphorylation followed by two successive rounds of reduction to yield DHB (Fig. 1). The negative standard Gibbs free energy for the pathway (Supplementary Note 1; Supplementary Table 1) attests to its thermodynamic viability. Stoichiometric analysis of the metabolic network in E. coli shows that DHB can be produced from glucose with a theoretical maximum yield of $1.5 \, mol \, mol^{-1}$ (Supplementary Note 2; Supplementary Fig. 1). Given that DHB can be converted into the methionine-analogue HMTB without carbon loss, the production of methionine via DHB increases the theoretical yield by ∼100% compared to the conventional one-step biosynthesis of methionine from glucose and sulfate, and by ∼30% when compared to the biosynthesis of methionine from glucose and methanethiol[10].

Given the structural similarity of DHB and homoserine pathway intermediates, we identified three homoserine pathway enzymes as prototype candidates for screening and engineering of enzyme activity in the DHB pathway. We chose AK, encoded by lysC[17], and ASD, encoded by asd[18], from E. coli, and HSD, encoded by HOM6 (ref. 19), from Saccharomyces cerevisiae since the HSD enzymes in E. coli are bifunctional enzymes with an associated AK activity[20,21]. We found that Ec-Asd had only trace activity on the synthetic substrate ($k_{cat} = 0.13 \, s^{-1}$ on malyl-P versus $36 \, s^{-1}$ on aspartyl-4-phosphate (aspartyl-P), whereas Ec-LysC and Sc-Hom6 had no detectable activity on malate and malate semialdehyde, respectively (Supplementary Table 2; Supplementary Note 3). These findings indicated that the feasibility of the envisaged de novo synthetic pathway necessitated the engineering of all three enzyme activities.

**Engineering of malate kinase activity.** We first set out to engineer MK activity into aspartate kinase III from E. coli (Ec-LysC). Binding interactions of the (L)-aspartate natural substrate in the enzyme active site are revealed in the 2.5 Å X-ray crystallographic structure of the dimeric wild-type LysC abortive ternary complex with (L)-aspartate and the Mg-ADP reaction co-product in the R-state[22]. The amino-acid substrate is anchored in position via water-mediated interactions of the β-carboxylate group with the metal ion in Mg-ADP, and by respective salt bridge electrostatic interactions of the (L)-aspartate α-amino and α-carboxylate groups with charged enzyme side-chain functional groups of Glu119 and Arg198 as shown in Supplementary Fig. 2. Productive binding of the (L)-aspartate substrate is further supported by a network of van der Waals and hydrogen bonding interactions with other residues in the active site.

(L)-Malate is an isostere of (L)-aspartate in which the positively charged α-amino group is replaced by an uncharged hydroxyl group. In common with other (succinate and malonate) structural analogues of (L)-aspartate that do not contain an α-amino group, and which therefore cannot form a salt-bridge with the side-chain of Glu119, (L)-malate has been reported to be a weak competitive inhibitor of Ec-LysC with a $K_i$ of 53 mM (ref. 23). These observations suggest that (L)-malate binds non-productively to the wild-type enzyme in the absence of additional binding energy provided by the salt-bridge interaction.

Figure 1 | The synthetic 2,4-dihydroxybutyrate (DHB) pathway is inspired by the natural homoserine pathway.

To investigate the impact of Glu119 substitution by other amino acids, we carried out saturation mutagenesis at this residue position and analysed the enzymatic activities of the resulting mutants. Trace but measurable enzymatic activity on (L)-malate could be detected with the Ec-LysC E119Q and Ec-LysC E119N mutants, in which the Glu119 side-chain carboxylate group is replaced by an uncharged carbamoyl group (Supplementary Table 3). Significant increases of up to 200-fold in the $k_{cat}/K_m$ value for (L)-malate were obtained in E119G, E119A, E119S and E119C variants with shorter side-chains at residue position 119 (Fig. 2a; Supplementary Table 3).

To further improve the catalytic efficiency towards (L)-malate, the construction of a small combinatorial mutant library was undertaken. Analysis of binding interactions of Ec-LysC in the experimental complex with (L)-aspartate enabled the selection of eight amino-acid residue positions including Glu 119, highlighted in Supplementary Fig. 2. Of these eight positions, five make direct contact with the (L)-aspartate substrate (Ala40, Thr45, Glu119, Phe184 and Ser201), and three others (Val115, Thr195 and Thr359) are located within a second residue shell not in direct contact with the substrate. The design of the library, described in the Supplementary Note 4, took account of natural sequence variation in the active-site region of Ec-LysC homologues and the results of computational re-design at nine residue positions in the vicinity of residue position 119. Restrictions placed on the number of permitted mutations at each of the eight residue positions in the library constrained its overall size to 2,160 possible theoretical combinations (Supplementary Table 5). A miniaturized screening protocol (see Methods) was devised to permit the direct measurement of (L)-malate kinase activity by a single end-point measurement with an accuracy of ± 8% in microtiter plates. To ensure adequate sequence space coverage[24], 6,720 clones were tested by this method leading to the identification of nine positive variants which were confirmed by enzymatic assay of sequenced and individually purified clones. The best mutant Ec-LysC V115A:E119S:E434V exhibited a $k_{cat}/K_m$ value of $0.82 \, s^{-1} \, mM^{-1}$ on (L)-malate, approximately only twofold lower than that of the wild-type enzyme acting on

the (L)-aspartate natural substrate. In addition, the mutant retained very little activity towards (L)-aspartate, resulting in a marked change in enzyme specificity (Fig. 2b). It is of note that the Glu434 position at the enzyme surface was not targeted in the construction of the library, but an E434V mutation was unexpectedly found in the best malate kinase mutant. A molecular model of (L)-malate bound in the active-site of the ternary complex with the Ec-LysC V115A:E119S double mutant and Mg-ADP is shown in Fig. 2c.

To render the malate kinase enzyme more efficient for *in vivo* applications, we individually tested E250K, T344M, S345L and T352I mutations previously shown to alleviate feedback inhibition by lysine in the wild-type enzyme[25]. We found that all these mutations strongly decreased the inhibitory effect of lysine on malate kinase activity (Supplementary Fig. 4). The quadruple mutant, Ec-LysC V115A:E119S:E250K:E434V was therefore selected for implementation of the DHB pathway.

**Engineering of malate semialdehyde dehydrogenase activity.** The aspartate semialdehyde dehydrogenase from *E. coli* (Ec-Asd) enzyme was found to possess only trace activity on malyl-P in the reductive (biosynthetic) reaction direction, and on malate semialdehyde (MSA) in the reverse oxidative phosphorylation reaction (Supplementary Table 6). We first sought to engineer increased activity towards the malyl-P/MSA substrate/product couple through site-directed mutagenesis of the Ec-Asd Gram-negative bacterial enzyme. Active site residues involved in the binding of aspartate semialdehyde (ASA) to Ec-Asd have been previously identified in an X-ray crystal structure of a covalent complex formed as the reaction product of Cys135 thiol group attack on the substrate analogue S-methylcysteine sulfoxide in the presence of NADP$^+$ (ref. 26). The binding of the ASA α-amino and α-carboxylate groups in the Ec-Asd active-site occurs via salt-bridge interactions with oppositely charged Glu241 and Arg267 residue side-chains. This salt bridging arrangement is similar to that of the (L)-aspartate substrate α-amino and α-carboxylate groups in the complex with *E. coli* aspartate

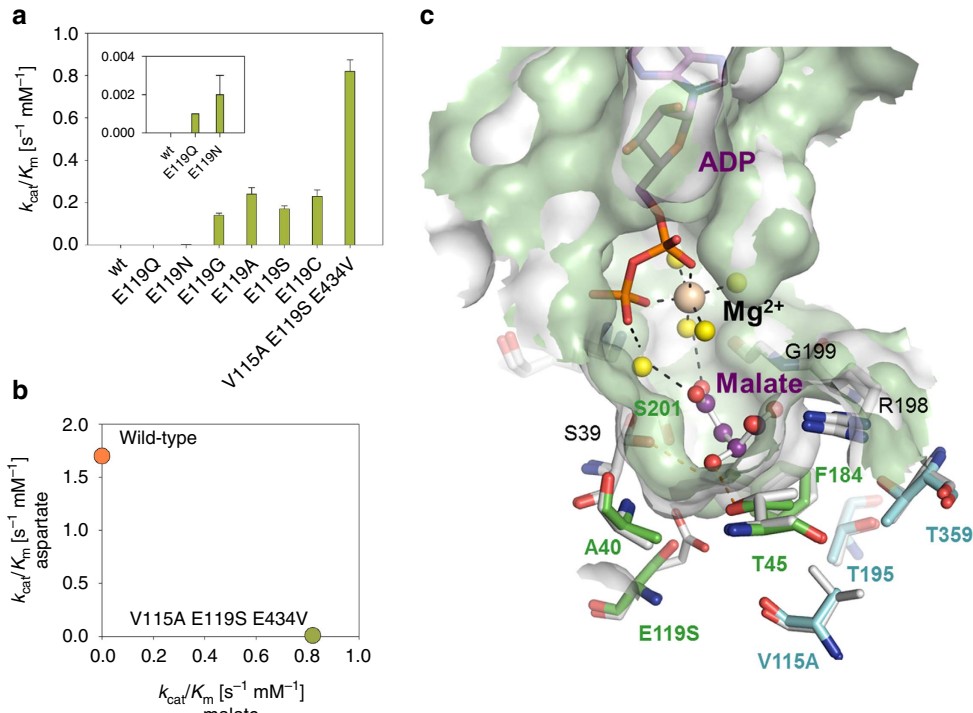

**Figure 2 | Engineering of malate kinase activity.** (**a**) Catalytic efficiency ($k_{cat}/K_m$) of wild-type aspartate kinase and the best malate kinase mutants on (L)-aspartate and (L)-malate. Kinetic data are from Supplementary Table 3. The results are the mean of at least two biological replicate experiments. Error bars correspond to the standard deviation of the mean. (**b**) Catalytic efficiencies ($k_{cat}/K_m$) of wild type LysC and the V115A E119S E434V Lys C triple mutant on (L)-aspartate and (L)-malate. Kinetic data are from Supplementary Table 3. (**c**) Active-site region in molecular model of complex of the *E. coli* (Ec-LysC) E119S:V115A double mutant with (L)-malate and Mg-ADP. Carbon atoms in ADP and (L)-malate are coloured in purple. The $Mg^{2+}$ ion is depicted as an ochre-coloured space filling sphere, and water molecules mediating binding interactions as yellow spheres. Enzyme residue positions in the combinatorial library for experimental screening of malate kinase activity (Supplementary Table 5) are highlighted with green (or cyan) coloured carbon atoms according to whether (or not) direct residue contact can be made with (L)-malate in the model complex. Atoms are otherwise coloured according to element type: other carbon, grey; nitrogen, blue; oxygen, red; and phosphorus, orange. Hydrogen bond interactions are shown as dashed-line vectors connecting donor and acceptor heavy atom positions. The model is overlaid on the X-ray structure of the R-state wild-type enzyme complex with (L)-aspartate and Mg-ADP (PDB code 2j0w) from which it was derived as described in Methods. Diffuse molecular surface representations of the mutant and wild-type enzyme active-sites are respectively shown in green and grey.

kinase III (ref. 22) that catalyses the preceding reaction step in the physiological pathway. The 2-OH group in a malyl-P/MSA substrate/product couple might be expected to hydrogen bond with Glu241 in Ec-Asd thereby providing for substrate binding similar to that of the natural substrate derivative in the experimental complex. However, the poor observed activity of wild-type Ec-Asd on MSA compared to ASA may be in part due to a lowering in the binding affinity for an alternative substrate carrying net negative charge. Replacement of the conserved Glu241 residue in the wild-type *E. coli* enzyme by residues with uncharged side-chains would then be expected to improve MSA binding affinity and reduce that of ASA.

To test this hypothesis, saturation mutagenesis of Ec-Asd was carried out at residue position 241. Since aspartyl-P and malyl-P are highly unstable molecules, initial kinetic characterization of the wild-type and mutant enzymes was carried out on the MSA substrate of the reverse (oxidative phosphorylation) reaction. The wild-type enzyme and the most promising mutants (E241Q and E241C, see Supplementary Fig. 5) were then kinetically characterized in the biosynthetic (physiological) reaction direction in coupled enzyme reaction assays, using MK or AK to generate an *in situ* supply of malyl-P or aspartyl-P as appropriate. We found that the introduction of mutations E241Q and E241C respectively improved enzyme specificity by 71- and 17-fold in favour of the malyl-P substrate (Fig. 3a). However, the

change in specificity was brought about by a marked decrease in activity towards the natural substrate aspartyl-P, rather than an intrinsic increase in activity towards malyl-P (Fig. 3b; Supplementary Table 6). This result showed on one hand that the electrostatic interaction between the negatively charged Glu241 and the positively charged amino group of the natural aspartyl-P substrate was a major requirement for the wild-type activity of this enzyme. On the other hand, it became clear that MSD activity in Ec-Asd could not be significantly increased by simply imposing more favourable polar interactions between the α-hydroxyl group of malyl-P and alternative amino-acid residues in position 241. The active-site region in a modelled structure of a putative hemithioacetal MSA substrate derivative covalently bound to Cys135 in the Ec-Asd E241Q mutant (see Methods) is shown in Fig. 3c. The figure highlights electrostatic interactions between residues in the mutant enzyme active site and the covalently bound MSA reaction intermediate.

In an attempt to further increase malyl-P reductive dephosphorylation activity, orthologues of the ASD family from the Gram negative, and archaeal and fungal phylogenetic branches were examined as alternative enzyme engineering platforms. ASDs from these phylogenetic branches are differentiated by the presence of characteristic structural insertions and deletions in the co-enzyme binding-site region and at the enzyme homodimer subunit interface. Although a high degree of conservation of

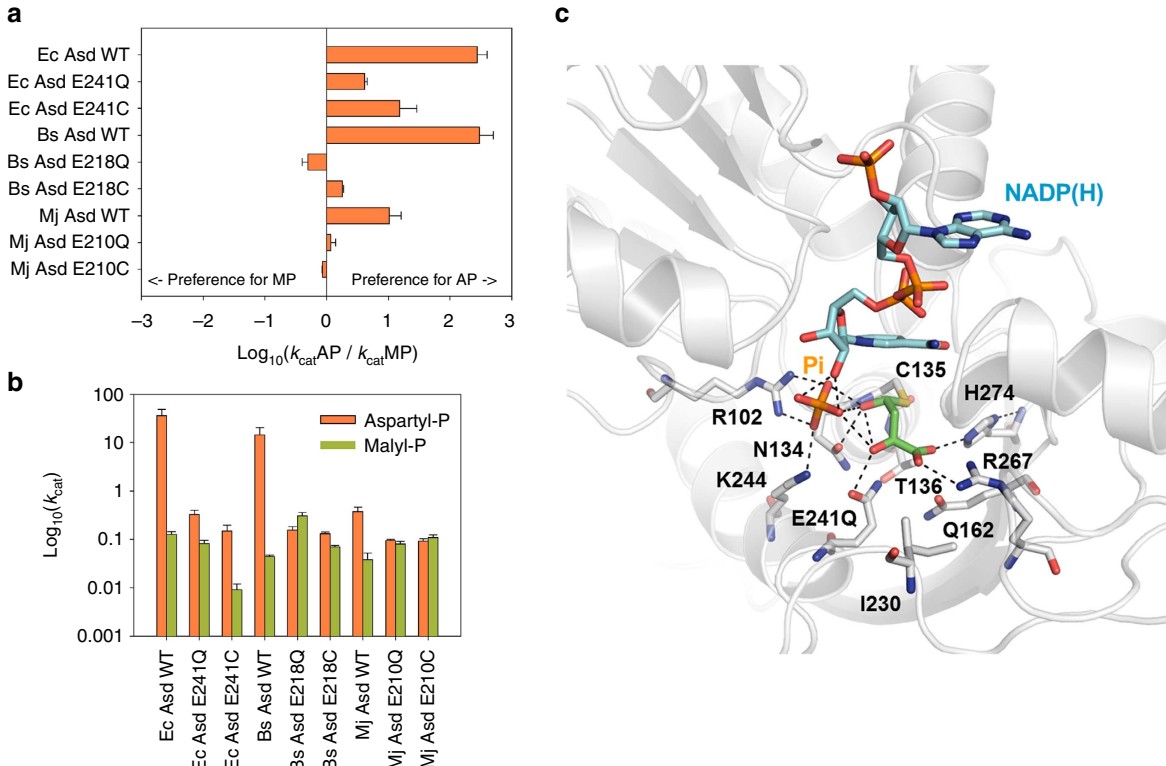

**Figure 3 | Engineering of malate semialdehyde dehydrogenase activity.** (**a**) Specificity (expressed as the ratio of maximum activity on aspartyl-P and malyl-P) of wild-type and mutant ASD enzymes from *E. coli* (Ec), *B. subtilis* (Bs), and *M. jannaschii* (Mj). (**b**) Activity of these enzymes on aspartyl-P and malyl-P. The results are the mean of at least two biological replicate experiments. Error bars correspond to the standard deviation of the mean. (**c**) Putative hemithioacetal malate semialdehyde (MSA) tetrahedral covalent reaction intermediate attached to Cys135 in the active-site of a computer-built model of an E241Q mutant of *E. coli* aspartate semialdehyde dehydrogenase (Ec-Asd) quaternary complex with non-covalently bound NADP(H) co-enzyme and inorganic phosphate (Pi). Carbon atoms in stick representations of enzyme residues, MSA reaction intermediate and the co-enzyme are respectively coloured in grey, green and cyan. Other atoms are shown in blue (nitrogen), red (oxygen), yellow (sulfur) and orange (phosphorus). Hydrogen bond interactions are represented as dashed-line vectors.

amino-acid residue functional groups exists within the active-site core, shared sequence identities of ASD family enzymes fall to as low as 10% (ref. 27). This structural variation correlates closely with marked differences in ASA oxidative phosphorylation catalytic efficiency, which varies by two orders of magnitude[28]. The natural variation in structure and sequence afforded by Gram-positive bacterial and *archaeal* ASDs may thus provide opportunities to modulate kinetic reaction rates other than through the introduction of additional mutations in the enzyme active-site.

The activities of wild-type and mutant ASDs from the Gram-positive bacterium *Bacillus subtilis* (Bs), that shares 26% sequence identity with Ec-Asd, and the *archaeon Methanocaldococcus jannaschii* (Mj) which is 21% identical to Ec-Asd, were assayed on aspartyl-P and malyl-P. Enzymatic activities of wild-type Bs-Asd and Mj-Asd on malyl-P were approximately threefold lower than those observed for Ec-Asd (Fig. 3b; Supplementary Table 6). However, by analogy to the effects observed in Ec-Asd the reaction specificity of these enzymes is significantly improved in favour of malyl-P when the conserved active-site glutamate residues (Glu218 in Bs-Asd, Glu210 in Mj-Asd) are mutated to glutamine or cysteine. The best result was obtained for the Bs-Asd E218Q mutant enzyme, which displayed an almost eightfold increased activity on malyl-P, and a $k_{cat}$ value of 0.31 s$^{-1}$. This is comparable with Mj-Asd wild-type activity towards aspartyl-P ($k_{cat} = 0.38$ s$^{-1}$). It is, however, much lower than that of the Bs-Asd wild-type ($k_{cat} = 14.5$ s$^{-1}$). The kinetically efficient

alternating-sites mechanism operating in bacterial ASD forms is absent in Mj-ASD and other *archaeal* ASDs[28,29]. Comparison of the relative turnover numbers suggests that the conversion of malyl-P to MSA catalysed by the Bs-Asd E218Q mutant may also be rate-limited by decoupled inter-subunit communication. Taking into account the loss in activity on the natural substrate, the specificity of Bs-Asd E218Q was shifted 650-fold towards malyl-P (Fig. 3a,b; Supplementary Table 6). This enzyme variant was therefore chosen for integration in the construction of the DHB pathway.

**Engineering of malate semialdehyde reductase activity.** Having found homoserine dehydrogenase from *Saccharomyces cerevisiae* (Sc-Hom6) to be inactive on MSA, we sought to identify an alternative enzyme template for the engineering of MSR activity. We experimentally screened for MSA reductase activity in oxidoreductases that act on substrates structurally similar to MSA (Supplementary Table 7). Significant MSA reductase activity was detected for the broad-range aldehyde reductase, Ec-YqhD[30], from *E. coli*, methylbutyraldehyde reductase, Sc-Ypr1 (ref. 31), from *S. cerevisiae*, 4-hydroxybutyrate dehydrogenase, Pg-4hbd[32,33], from *Porphyromonas gingivalis*, and the succinic semialdehyde dehydrogenase, Ms-Ssr[34], from *Metallosphaera sedula*. Amongst the enzymes that were active on MSA, the NADP$^{+}$-dependent succinic semialdehyde reductase from *Metallosphaera sedula*[34] (Ms-Ssr) displayed the highest specific

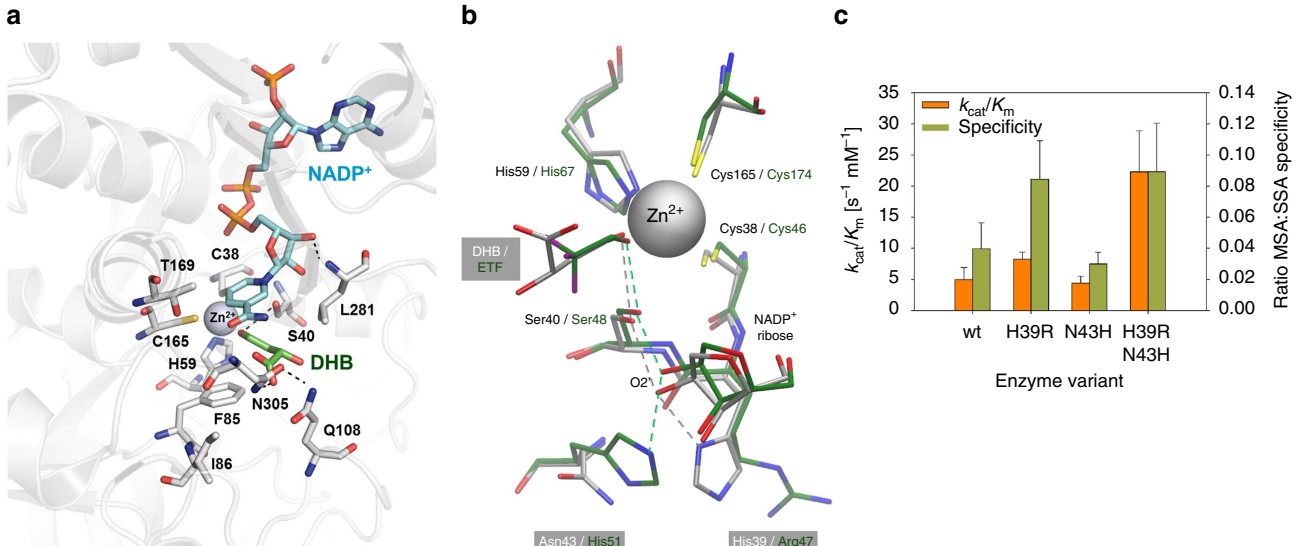

**Figure 4 | Engineering of malate semialdehyde reductase activity.** (**a**) Active-site region in a modelled complex of zinc-bound alcoholate ion of 2,4-dihydroxybutyric acid (DHB) with *M. sedula* succinic semialdehyde reductase (Ms-Ssr) and NADP$^+$ co-enzyme. Carbon atoms in stick representations of enzyme residues, DHB and the co-enzyme are respectively coloured in grey, green and cyan. Other atoms are coloured according to element type: nitrogen, blue; oxygen, red; sulfur, yellow; phosphorus orange. Dashed lines indicate hydrogen bond interactions. The Zn$^{2+}$ ion is shown as a space filling sphere in grey. (**b**) Alternative proton relay systems operating in horse liver alcohol dehydrogenase (ADH1) and Ms-Ssr enzyme homologues. The overlay shows topologically equivalent residue positions in the modelled complex of Ms-Ssr and the zinc-bound DHB alcoholate ion (grey coloured carbon atoms) and the X-ray crystal structure of the horse liver ADH1 F93W mutant ternary complex (PDB code 1axe) with NAD$^+$ and trifluoroethanol (ETF) inhibitor (carbon atoms in green). Proton relay shuttles in the two enzymes are depicted as correspondingly grey and green coloured inter-atomic dashed-line connecting vectors. Other atoms are coloured according to element type as in (**a**), with fluorine atoms in ETF additionally shown in purple. The Ms-Ssr wild-type proton relay can be interchanged with the archetypal ADH1 shuttle in the Ms-Ssr H39R:N43H double mutant. (**c**) Kinetic parameters for Ms-Ssr mutants on succinic (SSA) and malic (MSA) semialdehyde. The results are the mean of at least three replicate experiments. Error bars correspond to the s.d.

activity (4.0 µmol min$^{-1}$ mg$^{-1}$) and the best affinity (1.1 mM). However, its $k_{cat}/K_m$ value was 112-fold lower than that of the (Sc-Hom6) homoserine dehydrogenase benchmark enzyme in the natural pathway on aspartate semialdehyde (Supplementary Table 7; Supplementary Note 5). Therefore, a comparative molecular modelling approach based on the exploitation of conserved relational structural and functional features was used to improve the catalytic efficiency of Ms-Ssr towards MSA.

Ms-Ssr is a zinc-dependent alcohol dehydrogenase[34] belonging to the medium-chain dehydrogenase/reductase (MDR) alcohol dehydrogenase (ADH) superfamily[35]. A molecular model of the Ms-Ssr dimer in a ternary complex with NADP$^+$ and (L)-2,4-dihydroxybutyrate (DHB) was constructed using the atomic co-ordinate data from experimentally determined alcohol dehydrogenase structures (see Methods). Binding interactions of the zinc-coordinated DHB alcoholate anion in the active-site region are shown in Fig. 4a. The main contacts of the DHB α-carboxyl and 2-hydroxyl groups in the model are with the Gln108 and Asn305 residues. The presence of bulky Phe85 and Leu281 side-chains lining the substrate binding pocket suggests that the Ms-Ssr enzyme has an intrinsic preference for (unbranched) primary alcohol substrates. Structural analysis furthermore revealed that the enzyme employs a proton relay pathway which is different from the archetypal pathway that is present in most of the ADH family enzymes, and which is considered to be less efficient[35–37] (Fig. 4b). The engineering of an ADH1E-like proton relay system into Ms-Ssr, comprising a histidine at position 43 and an arginine at position 39, was thus considered to be a promising means of directly improving Ms-Ssr catalytic activity.

The Ms-Ssr H39R:N43H double mutant exhibited an approximately fourfold increased catalytic efficiency on MSA compared to the wild-type enzyme, but still retained a strong preference for SSA over MSA (Fig. 4c, Supplementary Table 8). However, SSA is produced in *E. coli* only under extreme acid stress conditions, and its supply can be turned off by deleting both glutamate decarboxylases GadA and GadB[38]. Potential competition between SSA and MSA was therefore not considered to be a significant impediment to the efficient biosynthesis of DHB, and the Ms-Ssr H39R:N43H enzyme was thus chosen for incorporation in the DHB pathway.

**Minimal metabolic engineering enables biosynthesis of DHB.** To assure functional expression of the DHB pathway, we assembled a synthetic operon in the medium copy number pACT3 plasmid[39], which expressed the three genes encoding the best DHB pathway enzymes (MK: Ec-LysC V115A:E119S:E250K:E434V, MSD: Bs-Asd E218Q, MSR: Ms-Ssr H39R:N43H) under the control of the inducible tac promoter. The resulting plasmid pDHB was transformed into the wild-type *E. coli* MG1655 strain and produced 60 mg l$^{-1}$ DHB after 24 h of shake-flask cultivation in glucose mineral medium. The wild-type control strain that expressed the corresponding homoserine pathway enzymes (lysine resistant AK: LysC E250K, ASD: Ec-Asd, HSD: Sc-Hom6) produced no detectable quantities of DHB (Fig. 5; Supplementary Table 9). These results indicated that cellular DHB production via the synthetic pathway can be achieved, and we set-out to increase DHB production by metabolic engineering of the host strain and by optimization of the expression system.

The DHB-producing strain accumulated 50% more acetate than the control strain (Fig. 5). Therefore, we first tested whether DHB production could be increased by inactivating the pyruvate

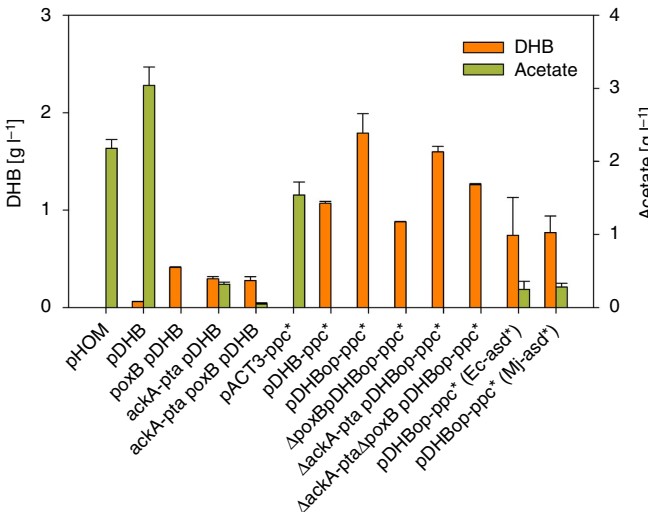

**Figure 5 | Production of 2,4-dihydroxybutyrate by engineered *E. coli* strains.** Cells were cultivated in shake flasks on mineral medium containing 20 g per l glucose. Values correspond to concentrations after 24 h of cultivation. Error bars represent STDV from at least two replicate experiments. All strains were derived from *E. coli* K-12 substr. MG1655. All plasmids were derived from the pACT3 medium-copy number plasmid. pHOM expresses the genes encoding the homoserine pathway enzymes AK: Ec-LysC E250K, ASD: Ec-Asd, HSD: Sc-Hom6. pDHB expresses the genes encoding the DHB pathway enzymes MK: Ec-LysC V115A:E119S:E250K:E434V, MSD: Bs-Asd E218Q, MSR: Ms-Ssr H39R:N43H. pDHB-ppc* additionally expresses the malate-insensitive PEP carboxylase mutant Ppc$_{K620S}$. pDHBopt-ppc* has optimized ribosome binding sites in front of each DHB pathway gene. pDHBopt-ppc*(Ec-asd*) and pDHBopt-ppc*(Mj-asd*) express, respectively, the Ec-Asd$_{E241Q}$ or Mj-Asd$_{E210Q}$ mutant enzymes instead of Bs-Asd$_{E218Q}$.

oxidase (PoxB)[40], or the acetate kinase/phosphate acetyl-transferase (AckA-Pta)-dependent[41] acetate pathways. The deletion of *poxB* and *ackA-pta* alone or in combination significantly reduced the production of acetate (Fig. 5). DHB production after 24 h increased to 0.3 g l$^{-1}$ and 0.4 g l$^{-1}$ in the $\Delta$*ackA-pta* and *$\Delta$poxB* mutants, respectively. The deletion of both acetate pathways did not further improve strain performance and yielded 0.27 g l$^{-1}$ DHB (Fig. 5, Supplementary Table 9). Furthermore, conversion of malate into DHB by the synthetic pathway reduces the regeneration of oxaloacetate via the Krebs cycle. The resulting imbalance between oxaloacetate and acetyl-CoA production therefore may have led to the conversion of excess acetyl-CoA into acetate. To increase the availability of oxaloacetate, we overexpressed the anaplerotic phosphoenol-pyruvate carboxylase (Ppc) enzyme. To prevent inhibition of the Ppc activity by aspartate or malate[42], we employed the Ppc K620S mutant which is insensitive to feedback inhibition by the two compounds[43] and cloned the corresponding gene into the DHB operon resulting in plasmid pDHB-ppc*. When this plasmid was expressed in a wild-type strain, 1 l DHB was produced after 24 h of cultivation and only trace amounts of acetate were detectable (Fig. 5). This result lends weight to the idea that a stoichiometric imbalance between the production of oxaloacetate and acetyl-CoA was indeed the major cause of increased acetate production in cells expressing the pDHB plasmid.

To improve the expression of the DHB pathway enzymes, their individual ribosome binding sites (RBS) were optimized[44]. The operon with optimized RBS sequences was assembled by fusion PCR and cloned downstream of the *ppc$_{K620S}$* gene whose RBS was not changed since moderate overexpression of Ppc has been

previously reported to be more beneficial for the production of Krebs cycle-derived C4 acids than maximizing Ppc activity[45–47]. When the optimized pDHB$_{opt}$-ppc* plasmid was expressed in a wild-type strain, we observed the accumulation of 1.8 g l$^{-1}$ DHB after 24 h of cultivation, corresponding to a molar yield of 0.15 (Fig. 5; Supplementary Table 9). This nearly twofold improvement can be attributed to a significant increase of MK and MSD activities, whereas the Ppc and MSA reductase activities remained nearly unchanged upon expression of the new plasmid (Supplementary Table 10). Expression of the optimized plasmid in the acetate pathway mutants did not further increase DHB production (Supplementary Table 9), in accord with the absence of acetate accumulation in DHB-producing strains. Furthermore, DHB synthesis dropped to $\sim$0.7 g l$^{-1}$ when Mj-Asd E210Q or Ec-Asd E241Q were applied as the MSD enzyme, indicating that the superior *in vitro* performance of Bs-Asd E218Q also translates into the best cellular DHB production (Fig. 5). Taken together our results show that overexpression of the anaplerotic Ppc activity is a key factor in enabling cellular DHB production, and that increasing the barely measurable MSD activity is a major requirement for the further improvment of DHB production levels.

## Discussion

Biosynthesis of value-added chemicals is typically achieved by metabolic engineering, which relies on alleviating product feedback inhibition of natural metabolic pathways and optimizing cofactor supply to enable overproduction of a target molecule. However, many molecules of chemical interest, such as DHB, are not naturally produced by living cells and annotated metabolic pathways for their biosynthesis do not exist. The development of tools and methodologies for the construction of synthetic pathways that expands the range of biochemically produced compounds or that improves stoichiometric yields in the biosynthesis of natural metabolites is therefore of major interest[2,48]. However, only a handful of studies have so far succeeded in the creation of pathways for the production of unnatural metabolites, and these largely rely on the recombination of cognate enzymatic activities to build up the desired pathway[49]. While the concept of implementing synthetic pathways through enzyme engineering for the biosynthesis of unnatural metabolites has been formulated, and the industrial potential of this technology has been recognized[2,48,49], experimental studies that demonstrate the actual feasibility of fully synthetic enzyme pathway construction are still lacking.

Our study demonstrates that the implementation of an efficient multistep non-natural DHB pathway is possible by applying design principles of synthetic biology, that is, conceptual pathway design and thermodynamic evaluation, computer-aided enzyme development, and strain engineering, in a straightforward manner. We have built upon the pioneering work of Holbrook *et al.*[50] who were the first to show that structure-guided rational protein engineering can provide new enzymatic functions, and extended their ideas to the redesign of an entire fully synthetic enzymatic pathway. Our work shows that industrial biosynthesis of non-natural value-added chemicals through synthetic biology is a realizable objective.

The engineering of the individual enzymatic activities was crucial in achieving a significant level of DHB production. This is supported by the fact that the expression of the unmodified template enzymes did not yield any DHB. In line with this result, our group[51] and Li *et al.*[52] detected only trace amounts of DHB (6.4 mg l$^{-1}$) when investigating an alternative non-natural DHB pathway, which employs malyl-CoA synthetase, malyl-CoA reductase, and malate semialdehyde reductase activities, and

which was assembled using natural enzymes with only residual activities on the synthetic substrates.

The activities of the corresponding homoserine pathway enzymes may serve as a benchmark in a quantitative assessment of the synthetic activities obtained. The catalytic performance of the best malate kinase mutant (Ec-LysC V115A:E119S:E250-K:E434V) is only two-fold lower than the activity of the wild-type AK enzyme (LysC) on the natural substrate, and can therefore be considered adequate for the biosynthesis of DHB. In contrast, the best MSD (Bs-Asd E218Q) and MSR (Ms-Ssr H39R:N43H) enzymes have approximately 50- and 25-fold lower activities than their respective ASD and HSD (Bs-Asd, Sc-Hom6) counterparts. Thus, further improvement of these enzymatic activities is necessary to increase DHB production via our pathway.

The engineering of appropriate host strains is another important factor for achieving efficient product formation. In the present work we have shown that the expression of a malate-insensitive anaplerotic enzyme (Ppc K620S) significantly increases DHB production. It is expected that additional metabolic engineering to optimize both the supply of the natural DHB precursor malate and the required cofactors will have an additional positive impact on DHB production.

The biosynthesis of DHB from renewable carbon sources can enable a more sustainable production of methionine, whose annual production volume increases at an average rate of $\sim 5\%$, to satisfy the increasing demand for this amino acid in the production of animal protein[9]. In addition, DHB may serve as an intermediate for the (bio)chemical synthesis of other value-added products. Recently, synthetic metabolic pathways have been disclosed that start from DHB to produce 1,3-propanediol[53], or 1,2,4-butanetriol[52]. Furthermore, 2-hydroxy-γ-butyrolactone, which is spontaneously formed from DHB at acidic pH, can be chemically converted to γ-butyrolactone, and 1,4-butanediol[54]. All of these compounds are bulk chemicals with a wide range of applications as precursors for plastics, solvents and propellants[15,52,55]. Thus, we consider DHB to be a versatile platform molecule with high industrial potential. Our study describes an efficient biosynthetic route for the production of this hitherto inaccessible molecule from renewable resources.

## Methods

**DNA manipulations.** Restriction endonucleases and DNA-modifying enzymes were purchased from New England Biolabs and used according to manufacturer's instructions. DNA plasmid isolation was performed using GeneJET Plasmid Miniprep Kit (Thermo scientific). DNA extraction from agarose gel was carried out using the GeneJET Gel Extraction Kit (Thermo scientific). DNA sequencing was carried out by Beckman Coulter Genomics (Takeley, United Kingdom).

**Vectors for protein expression and purification.** Genes were amplified from genomic DNA (extracted from *Escherichia coli* MG1655, *Bacillus subtilis* BSB168 (kindly provided by Dr M. Jules, INRA, Paris), *Saccharomyces cerevisiae* BY4741, *Metallosphaera sedula* DSM5348 (kindly provided by Prof. G. Fuchs, University of Freiburg, Germany), using high fidelity DNA polymerase Phusion (Finnzymes) and primer pairs listed in Supplementary Table 11. The genes *asd* and *4hbd* from *Methanocaldococcus jannaschii* DSM2661 and *Porphyromonas gingivalis* W83, respectively, were synthesized by Eurofins. The resulting DNA fragments were digested with suitable restriction enzymes (Supplementary Table 11), cloned into the corresponding sites of pET28a (Novagen) using T4 DNA ligase (Biolabs), thereby adding an N-terminal hexa-His tag. The ligation product was transformed into *E. coli* DH5α cells (NEB) using standard protocols[56]. The resulting plasmids were isolated, and shown by DNA sequencing to contain correct sequences.

**Protein mutant construction by site-directed mutagenesis.** Site-directed mutagenesis was carried out on pET28-derived plasmids using the primer pairs listed in Supplementary Table 12. Point mutations to change the amino-acid sequences were introduced by PCR (Phusion 1U, HF buffer 20% (v/v), dNTPs 2.5 mM, direct and reverse primers 0.1–0.5 μM each, template plasmid $\sim 50$ ng, water to bring to 50 μl). When possible, plasmids created by PCR contained new restriction sites (introduced by silent mutations) in addition to the functional mutation to facilitate identification of mutated clones. The PCR products were digested by *DpnI* at 37 °C for 1 h to remove template DNA, and transformed into NEB 5-alpha competent *E. coli* cells (NEB). The mutated plasmids were identified by restriction site analysis and verified to carry the desired mutations by DNA sequencing.

**Protein expression and purification.** Expression of enzymes: *E. coli* BL21 (DE3) cells were transformed with the appropriate pET28a-derived plasmids using standard protocols[56]. Strains containing the expression vectors were pre-cultured overnight in Luria-Bertani (LB) medium before they were used to inoculate 50 ml of LB medium at $OD_{600} \sim 0.2$. Protein expression was induced by addition of 1 mM isopropyl β-D-1-thiogalactopyranoside (IPTG) when $OD_{600}$ reached $\sim 0.6$. After 3 h of incubation at 37 °C in LB containing 50 μg l$^{-1}$ kanamycin, cells were harvested by centrifugation at 13,000 × g for 10 min (Sorvall ST 40R, Thermo) and stored at $-20$ °C until further analysis.

Purification of enzymes: Frozen cell pellets from expression cultures were suspended in 0.5 ml of breakage buffer (50 mM Hepes, 300 mM NaCl, pH 7,5) and broken open by four successive rounds of sonication (Bioblock Scientific, VibraCell 72,437) with the power output set to 30%. Cell debris was removed by centrifuging the crude extracts for 15 min at 4 °C at 13,000 × g (Sorvall ST 40R, Thermo) and retaining the clear supernatant. RNA and DNA were removed from the extracts by adding 15 mg per ml streptomycin (Sigma), centrifuging the samples at 13,000 × g for 10 min at 4 °C and retaining the supernatant. Clear protein extract was incubated for 1 h at 4 °C with 0.75 ml bed volumes of Talon$^{TM}$ Cobalt affinity resin (Clontech). The suspension was centrifuged at 700 × g in a table top centrifuge (Sorvall ST 40R, Thermo) and supernatant was removed. The resin was washed with 10 bed volumes of wash buffer (50 mM Hepes, 300 mM NaCl, 15 mM Imidazole, pH 7,5) before enzymes were eluted with 0.5 ml of elution buffer (50 mM Hepes, 300 mM NaCl, 250 mM Imidazole, pH 7,5). In the case of aspartate kinase, and aspartate semialdehyde dehydrogenase preparations the elution buffer was replaced by wash buffer w/o imidazole using Amicon Ultra-0.5 ml-centrifugal filters (cut-off 10 kDa, 2 centrifugations at 13,000 × g for 7 min at 4 °C). Purity of eluted enzymes was verified by SDS-PAGE analysis. Protein concentrations were measured with the method of Bradford[57].

**Enzymatic assays.** Malate and aspartate kinase activity: Aspartate or malate kinase activities were assayed by coupling ADP production in the kinase reactions to NADH oxidation in the presence of phosphoenolpyruvate, pyruvate kinase, and lactate dehydrogenase. The assay mixture contained 50 mM Hepes (pH 7.5), 50 mM KCl, 5 mM MgCl$_2$, 0,24 mM NADH, 2 mM ATP, 1 mM PEP, 9 μg per ml of lactate dehydrogenase (Sigma, L2500), 12.4 μg per ml pyruvate kinase (Sigma, P1506), and appropriate amounts of purified aspartate (malate) kinase. Reactions were started by adding appropriate concentrations of (L)-aspartate or (L)-malate.

Malate and aspartate-β-semialdehyde dehydrogenase activity: Aspartate or malate-β-semialdehyde dehydrogenase activities were assayed in both the biosynthetic and the reverse sense of the reaction. Assays in the biosynthetic (reductive) direction were carried out by following oxidation of NADPH during the reduction of malylphosphate or aspartylphosphate. Since both substrates are unstable and not commercially available, they were produced *in situ* by the action of previously purified malate (Ec-LysC E119S) or aspartate (Ec-LysC) kinase. The assay mixture contained 50 mM Hepes (pH 7), 0.2 mM NADPH, 2 mM ATP, 5 mM MgCl$_2$ and 3 U per ml malate kinase or aspartate kinase. The reaction was started by adding 20 mM malate or aspartate. Assays in the reverse biosynthetic (oxidative) direction were carried out by following the reduction of NADP during the oxidation of aspartate semialdehyde or malate semialdehyde to 4-phospho-(L)-aspartate or 4-phospho-(L)-malate, respectively[58]. The assay mixture contained 200 mM Hepes (pH 9), 50 mM K$_2$HPO$_4$, 5 mM MgCl$_2$ and 0.25 mM NADP. Reactions were started by adding (L)-aspartate-β-semialdehyde (ASA) or (L)-malate-β-semialdehyde (MSA). ASA was added in the form of (L)-aspartic acid β-semialdehyde hydrate trifluoroacetate (maintained at pH3 to prevent degradation) which is a suitable substrate for enzymatic tests of homoserine dehydrogenase[58]. MSA was produced freshly prior to the enzymatic tests by the deprotection of the stable MSA derivative 2-[(4S)-2,2-dimethyl-5-oxo-1,3-dioxolan-4-yl]acetaldehyde(DMODA) (provided by Activation, France). Malate semialdehyde was obtained by boiling DMODA in water and evaporation of the released acetone (35 °C, 50 mbar). The pH of the malate semialdehyde solution was adjusted to 3 using sodium bicarbonate. It was verified that addition of the MSA solution did not change the pH of the final reaction mix.

Malate, aspartate and succinate semialdehyde reductase activity: The assay mixture contained 200 mM Hepes (pH 7.5), 50 mM KCl, 5 mM MgCl$_2$, 0.2 mM NADH or NADPH, and appropriate amounts of purified enzyme or cell extract. Reactions were started by adding appropriate amounts of malyl semialdehyde, aspartyl semialdehyde, or succinic semialdehyde. All enzymatic assays were carried out at 37 °C in 96-well flat bottomed microtiter plates in a final volume of 250 μl. The reactions were followed by the characteristic absorption of NADH at 340 nm in a microplate reader (BioTek EON).

**Construction of the aspartate kinase mutant libraries.** Mutagenesis of the lysC gene was carried out using the ISOR method[59]. Primers pETseq_for and pETseq_rev (Supplementary Table 13) were used to amplify the gene LysC from

the plasmid pET28-lysC. The purified PCR product was digested with DNaseI endonuclease to obtain fragments having an average size of ∼100 bp. Gel-purified fragments (∼100 ng in 30 μl reaction mix) were used for PCR-based gene reassembly using 2 μM of degenerated oligonucleotides that were designed to introduce the desired mutations (Supplementary Table 13) and Phusion High-Fidelity DNA Polymerase (Finnzyme). Full-length recombined genes were isolated from the reassembly products by nested PCR using the primers Ec_lysC_clon_for and Ec_lysC_clon_rev (Supplementary Table 13).

The PCR product was gel purified, digested with NdeI and BamHI and ligated into the corresponding sites of plasmid pET28a (Novagen). Commercial NEB 5-alpha competent *E. coli* cells were transformed with the ligation product and plated on LB agar supplemented with 50 μg per ml kanamycin. After overnight incubation at 37 °C colonies were scraped from the plates and plasmids were extracted. The DNA library was then transformed into chemo-competent *E. coli* BL21 Star (DE3) cells which were plated on LB agar plates containing 50 μg per ml kanamycin to obtain isolated colonies. A total number of 6,720 colonies was picked and transferred into 96-well microplates (Nunc Brands Products, Roskilde, Denmark) containing in each well 200 μl of LB medium supplemented with kanamycin and 8% glycerol. The storage microplates were kept at −80 °C after overnight culture at 30 °C.

**Screening of the aspartate kinase mutant library.** Screening of the enzyme variants was carried out using the automated robotics facilities of the ICEO platform (http://iceo.genotoul.fr/index.php?id=172&L=2). Storage microplates containing the *LysC* library were thawed and replicated to inoculate starter culture into 96-wells microplates (Nunc Brands Products, Roskilde, Denmark) filled with 200 μl 2x yeast extract- tryptone (YT) medium (16 g per l tryptone, 10 g per l yeast extract, 5 g per l NaCl) per well supplemented with kanamycin (50 μg per ml). The starter cultures were inoculated from the storage microplates and cultivated in 96-well microplates containing 200 μl of 2x YT medium and 50 μg per ml kanamycin. After 24 h incubation at 30 °C under horizontal shaking at 250 rpm (Infors HT, Bottmingen, Switzerland), 50 μl of each starter culture were used to inoculate the expression cultures which were cultivated in 96-deepwell plates (ABgene, Epsom, UK) containing 1 ml auto-inducing medium ZYM-5,052 (ref. 60) supplemented with kanamycin (50 μg per ml). Expression cultures were cultivated at 30 °C for 24 h shaken at 700 rpm in an incubator-shaker (INFORS HT, Bottmingen, Switzerland). The supernatant was removed by centrifugation (10 min, 3,700 × g, 4 °C) before the cell pellets were suspended in 200 μl of lysozyme solution (0.5 mg per ml), incubated for 30 min at 30 °C and stored at −80 °C for 12–24 h. Lysed cell pellets were thawed by incubation at room temperature for 1 h and 800 μl of benzonase solution (2.4 U per ml) were added to each well to digest DNA. Plates were incubated for 30 min at 30 °C before pelleting debris by centrifugation (10 min, 3,700 g, 4 °C). Protein extracts were diluted 250-fold in water by two successive transfers onto new microplates. 10 μl of the diluted enzymatic extracts were transferred onto a new microplate to which 65 μl of reaction mix (115 mM Hepes buffer pH 7.5, 5 mM ATP, 5 mM MgCl₂, 3.85 mM phosphoenolpyruvic acid, 2.88 mM NADH, 3.85 U per ml lactate dehydrogenase and 3.85 U per ml pyruvate kinase) was added. The reaction was started by adding 75 μl of 50 mM L-malic acid (neutralized with NaOH). Reactions were run for 40 min at 30 °C and stopped by adding 125 μl of 0.3 M HCl. After 20 min incubation at room temperature 60 μl of the reaction mix was transferred from each well onto a new microplate containing 190 μl of 9 M NaOH. Microplates were left in the dark during 3 h at room temperature to develop the alkali product from NAD⁺. The absorbance of each well was read at 360 nm using a BioTek Eon microplate spectrophotometer.

**Construction plasmids for biosynthesis of DHB.** All plasmids constructed and used in this study are listed in Supplementary Table 14.

Construction of pDHB operon: The three genes, *Ec-lysC*$_{V115A:E119S:E250K:E343V}$, *Bs-asd*$_{E218Q}$ and *Ms-ssr*$_{H39R\ N43H}$ were cloned into the medium-copy pACT3 vector[39] using the In-Fusion HD cloning kit (Clontech Laboratories, Inc.) and following the user manual instructions. Briefly, pACT3 was linearized by digestion with *Sac*I and *Hind*III restriction enzymes. Primer pairs lysC-IF-forwarder lysC-IF-reverse,asd-IF-forward/asd-IF-reverse and ssr-IF-forward/ssr-IF-reverse (Supplementary Table 15) were used to amplify *Ec-lysC*$_{V115A:E119S:E250K:E343V}$, *Bs-asd*$_{E218Q}$ and *Ms-Ssr*$_{H39R\ N43H}$ from pET28-*Ec-lysC*$_{V115A:E119S:E250K:E343V}$, pET28-*Bs-asd*$_{E218Q}$ and pET28-*Ms-Ssr*$_{H39R:N43H}$, respectively, using Phusion polymerase (Thermo-Scientific). The forward primer inserted a unique restriction site and RBS in front of the gene of interest, and the reverse primer had a 15 bp overlapping sequence with the next forward primer. lysC-IF-forward and ssr-IF-reverse primers had 15 bp of sequence complementarity with the 5′ and 3′ ends of the plasmid, respectively. The PCR products and the digested vector were assembled by homologous recombination using the reaction mix provided in the kit. The resulting plasmid was transformed into NEB 5-alpha competent *E. coli* cells (NEB) and verified by DNA sequencing to contain the correctly assembled operon.

Construction of pACT3-ppc*: The wild-type *ppc* gene was PCR amplified from genomic DNA of the *E. coli* K-12 substr. MG1655 (ATCC 47076) strain using Phusion polymerase (Thermo-Scientific) and the forward and reverse primers ppc_clon_for and ppc_clon_rev respectively (Supplementary Table 15). The

resulting DNA fragment was ligated into pACT3 using *Sma*I and *Xba*I restriction sites to obtain vector pACT3-ppc. The amino-acid exchange Lys620Ser was introduced into *ppc* by site directed PCR mutagenesis using primers ppc_k620s_for and ppc_k620s_rev (Supplementary Table 15). The PCR product was digested with *Dpn*I and transformed into NEB 5-alpha competent *E. coli* cells (NEB). The resulting plasmid, pACT3-ppc* was isolated and verified by DNA sequencing to contain the desired mutation.

Construction of pDHB-ppc*: The operon comprised of *Ec-lysC*$_{V115A:E119S:E250K:E343V}$, *Bs-asd*$_{E218Q}$ and *Ms-Ssr*$_{H39R\ N43H}$ was amplified from pDHB using the primers IF_ppc_lysC_For and IF_pACT_ssr_Rev (Supplementary Table 15). The resulting PCR fragment was cloned into the *Xba*I/*Hind*III-digested pACT3-ppc* plasmid using the In-Fusion HD cloning kit (Clontech Laboratories, Inc.). The resulting plasmid was transformed into NEB 5-alpha competent *E. coli* cells (NEB) and verified by DNA sequencing to contain the correctly assembled operon.

Construction of pDHBop-ppc*: The intergenic regions (including the ribosome binding sites (RBS)) of the DHB operon were optimized using the RBS calculator[61]. The optimized intergenic regions were introduced into the forward and reverse primers (Supplementary Table 15) that served to amplify the DHB pathway genes individually using pDHB as the matrix. The resulting PCR fragments were purified (GeneJet Gel Extraction Kit, Thermo) and used in two successive rounds of overlap PCRs[62] to assemble a DNA fragment which contained all three genes, and which was ligated into the XbaI/XhoI digested vector pACT3-ppc*. The resulting plasmid was transformed into NEB 5-alpha competent *E. coli* cells (NEB) and verified by DNA sequencing to contain the correctly assembled operon.

Construction of pDHBop(Mj-asd*)-ppc* and pDHBop(Ec-asd*)-ppc*: The *Mj-asd*$_{E210Q}$ gene was PCR amplified from pET28-*Mj-asd*$_{E210Q}$ using Phusion polymerase (Thermo-Scientific) and the forward and reverse primers Mj_asd_clon_for and Mj_asd_clon_rev (Supplementary Table 15), respectively. The forward primer contained an RBS sequence which had been optimized by the RBS calculator web-tool. The resulting DNA fragment was ligated into pDHBop-ppc* using SpeI and BglII restriction sites. The resulting plasmid was transformed into NEB 5-alpha competent *E. coli* cells (NEB) and verified by DNA sequencing. pDHBop(Ec-asd*)-ppc* was constructed analogously using the primers Ec_asd_clon_for and Ec_asd_clon_rev (Supplementary Table 15).

**Construction of strains for biosynthesis of DHB.** *Escherichia coli* K-12 substr. MG1655 (ATCC 47076) was used as the parental strain for the construction of DHB-producing strains. Deletion of the *ackA-pta* operon was carried out by using the method of Datsenko & Wanner[63]. The kanamycin resistance cassette of pKD4 was amplified using primers that had ∼50 bp homology to the flanking region of the genomic target locus (Supplementary Table 16).The PCR product was transformed into the target strain which expressed λ-Red recombinase from the pKD46 plasmid.

Kanamycin-resistant clones were selected on LB agar plates containing 25 μg per mlof the antibiotic and were verified by PCR. Deletion of *poxB* was achieved using the phage transduction method adapted from Miller[64]. The phage lysate was prepared from the *ΔpoxB* strain of KEIO collection[65]. Positive clones were selected on kanamycin-containing LB agar plates and verified by PCR analysis. The *kan* cassette was removed from the genome by expressing FLP recombinase from the pCP20 plasmid[66] and correct excision of the cassette was verified by PCR using locus specific primers (Supplementary Table 16). Plasmids were transformed into target *E. coli* strains using standard protocols[56] obtaining the strains listed in Supplementary Table 17.

**Cultivation conditions for biosynthesis of DHB.** All cultivations were carried out at 37 °C on an Infors rotary shaker running at 170 r.p.m. Precultures (3 ml LB medium in test tubes) were inoculated from glycerol stocks, cultivated overnight, and used to inoculate a second preculture (25 ml mineral medium in 250 ml shake flask) at an OD$_{600}$ of ∼0.1. Exponentially growing cells of this preculture were used to inoculate the production culture (25 ml mineral medium in 250 ml baffled shake flasks) at OD$_{600}$ of ∼0.05. IPTG was added at a concentration of 1 mmol$^{-1}$ when OD$_{600}$in the production cultures reached ∼0.6. One liter mineral medium contained: 20 g glucose, 18 g Na₂HPO₄*12 H₂O, 3 g KH₂PO₄, 0.5 g NaCl, 2 g NH₄Cl, 0.5 g MgSO₄*7 H₂O, 0.015 CaCl₂*2 H₂O, 1 ml of 0.06 molper l FeCl₃ stock solution prepared in 100 times diluted concentrated HCl, 2 ml of 10 mM thiamine HCl stock solution, 20 g MOPS, (and the antibiotics kanamycin sulfate and chloramphenicol at 50 μg l$^{-1}$ and 25 μg l$^{-1}$, respectively, when necessary), and 1 ml of trace element solution (containing per liter: 0.04 g Na₂EDTA*2H₂O, 0.18 g CoCl₂*6 H₂O, ZnSO₄*7 H₂O, 0.04 g Na₂MoO₄*2 H₂O, 0.01 g H₃BO₃, 0.12 g MnSO₄*H₂O, 0.12 g CuCl₂*H₂O). The pH was adjusted to 7 and the medium was filter-sterilized.

**Analytical methods.** The concentration of DHB was determined on a Dionex Ultimate 3,000 HPLC system (Thermo scientific, France) using an Aminex HPX-87H column (protected by a 'Micro-Guard cation H Refill Cartridge' precolumn) heated at 48 °C. Mobile phase was 0.01% (v/v) formic acid at 0.4 ml min$^{-1}$ isocratic flow rate. Depending on the concentration range, detection and quantification of DHB were performed by a dual-wave UV–vis (200 and 210 nm) detector (Dionex Diode Array Detector) or by a MSQ mass

spectrometer (Finnigan Surveyor MSQ plus) in electrospray mode (ESI probe at 0.25 kV) with negative ionization (ionization temperature, 450 °C; cone voltage, 50 volts). Glucose and organic acids were quantified on a Dionex Ultimate 3,000 HPLC system equipped with a UV and a RI (Shodex RI-101) detector using an Aminex HPX-87H column (protected by a 'Micro-Guard cation H Refill Cartridge' precolumn) heated at 35 °C, using 0.01% (v/v) sulfuric acid as mobile phase, at 0.5 ml min$^{-1}$ isocratic flow rate.

**Computational methods.** Molecular models of Ec-LysC wild-type and mutant complexes with (L)-malate were built by modification of the X-ray crystal structure of a ternary complex of the wild-type enzyme in the R-state with (L)-aspartate and Mg-ADP[22]. An all-atom starting set of co-ordinates was prepared by stepwise energy minimization of the experimental structure (PDB code 2j0w) with harmonic restraints initially placed on all heavy atom positions. Restraining forces on atom subsets comprising the (L)-malate ligand, mutated and neighbouring protein residues and water molecules were then removed before the release of all restraints in the final round of geometry refinement of the mutant enzyme model complexes. Potential energy force field parameter sets used in minimization are described below.

Computational re-design of the active-site region in Ec-LysC centred around residue position 119 was performed by the application of 30,000 Monte Carlo rounds of random mutagenesis at nine residue positions (Ser39, Ala40, Thr45, Val115, Glu119, Phe184, Thr195, Ser201 and Thr359) in a modelled binary complex of the E119Q mutant with (L)-malate using RosettaDesign software[67]. All possible exchanges with the 20 native amino-acid side-chains were permitted at each of the nine residue positions. Positions of all atoms in the (L)-malate ligand and in the protein main-chain were held fixed, along with side-chain atom positions in non-mutated residues.

A molecular model of the Ec-ASD E241Q mutant complex containing the hemithioacetal malate semialdehyde tetrahedral covalent reaction intermediate presumed to be formed, following phosphate release from malyl-P, by hydride transfer to a thioacyl-enzyme intermediate, was built from the template X-ray crystal structure (PDB code 1gl3) of the Ec-ASD covalent complex formed by reaction of the Cys135 thiol group with the S-methylcysteine sulfoxide substrate analogue inhibitor in the presence of NADP$^+$ (ref. 26). The covalently attached malate semialdehyde substrate derivative was constructed by superposition of a geometry-optimized structure of the model compound (4S)-4-{[(2S)-2-amino-3-oxo-propyl] sulfanyl}-L-malate in an analogous conformation to the L-cysteine analogue derivative in the crystallographic structure. An inorganic phosphate ion was docked into the vacant site occupied by the phosphate group of the malyl-P substrate by superposition of atomic co-ordinates (PDB code 1nx6) from the *Haemophilus influenza* ASD covalent complex with a hemithioacetal ASA partial reaction substrate derivative[68] at structurally aligned main-chain C$^\alpha$ positions. The dimeric structure was then energy minimized in a stage-wise manner with progressive release of harmonic force constraints on heavy atom positions.

An initial molecular model of the succinic semialdehyde dehydrogenase from *Metallosphaera sedula* (Ms-Ssr) was built from its amino-acid sequence (UniProtKB accession number A4YGN0) and the homologous (ADH1) horse liver alcohol dehydrogenase (PDB code 1axe) template structure of the F93W mutant complexed with NAD$^+$ and the inhibitor trifluoroethanol using the I-TASSER web server[69]. The shared sequence identity between Ms-Ssr and the model template was 39%. A NADP$^+$ coenzyme molecule was positioned by superposition at topologically equivalent C$^\alpha$ atomic co-ordinate positions of an ADH8 frog (amphibian) vertebrate X-ray crystallographic structure (PDB code 1p0f) with 37% sequence similarity to Ms-Ssr, containing the bound coenzyme, onto those of the model. Determination of the likely substrate binding mode was aided by the superposition of experimental structures of medium-chain dehydrogenase/reductase (MDR) ADH co-ordinator or substrate ligands chemically similar to the Ms-Ssr (L)-malate-4-semialdehyde substrate or (L)-2,4-dihydroxybutyrate reduction product (trifluoroethanol, 1axe; glycerol, 1p0f; cyclohexylfluoramide, 1e3i; 2-butanol, 1bxz; 1,4-butanediol, 2 × aa). A geometry-optimized structure of the 4-hydroxy-2-oxobutanoic acid alcoholate ion was manually docked into the binding site, guided by the atom positions of the overlaid ligand analogues. The resultant ternary complex was then energy minimized with harmonic positional restraints placed on all of the heavy atoms. A dimeric model structure was generated using the rotation matrix and translation vector for the superposition of the A and B chains in the 1axe co-ordinate data set.

A hydrogen-bonded proton relay pathway connecting the active-site with the bulk solvent exists in many MDR-ADHs to facilitate the transfer of the substrate hydroxyl proton to the exterior during the enzyme-catalysed reaction. The archetypal proton shuttle, first identified in the ADH1E (E-isoenzyme) of horse liver alcohol dehydrogenase[70], is composed of Ser48 and His51 (at the surface), bridged by the 2′-hydroxyl of the coenzyme ribose ring (Fig. 4b). However, His51 in the liver enzyme is replaced by an asparagine residue at the equivalent position (43) in Ms-Ssr. An alternative proton relay system[36] is believed to operate in Ms-Ssr in which the co-enzyme O2′ ribosyl hydroxyl oxygen atom instead hydrogen bonds with the side-chain of His39 (Fig. 4b). Visual inspection of the initial Ms-Ssr model, built using an ADH1 template structure in a 'closed' conformation[71], showed that the inter-atomic separation distance of 5.3 Å between the imidazole ring NE atom of His39 and the NADP$^+$ ribose O2′ was too long to

support the formation of a hydrogen bond. A structurally viable alternative proton relay system is observed in the X-ray structure (PDB code 1e3i) of the P47H mouse ADH2 mutant that exists in a 'semi-open' conformation in a ternary complex with NADH and cyclohexylformamide[37]. The P47H ADH2 mutant comprises an equivalent (His47/Asn51) residue pairing to that in Ms-Ssr (His39/Asn43), and the two enzymes possess an overall shared sequence identity of 37%. The structurally aligned helical region from Cys46 to Ala52 in the P47H ADH2 crystal structure was used as a fragment template for manual re-building of the rigid-body shifted equivalent (Cys38-Glu46) helical section in the initial Ms-Ssr model. Following a further round of restrained energy minimization of the refined monomeric structure, the Ms-Ssr dimer was regenerated as before, and subjected to a final round of energy minimization in the absence of harmonic restraints.

All energy minimizations were carried out using the *ff99SBAmber* molecular mechanics force field for protein atoms[72], and the GAFF force field[73] for atoms in ADP$^{3-}$, NADP$^+$, small molecule ligand substrates and their covalently attached derivative analogues. Water molecules present in parent wild-type experimental crystallographic structures were retained in the enzyme mutant molecular models. A distance-dependent dielectric constant was included in the Coulomb potential to allow for the solvent screening effect on interaction energies. Partial atomic charges for ADP$^{3-}$ and NADP$^+$ were taken from the CHARMM27 force field[74]. Partial charges for atoms in (L)-malate, (L)-2,4-dihydroxybutyrate (DHB), the negatively charged 4-hydroxy-2-oxobutanoic acid alcoholate ion and the (4S)-4-{[(2S)-2-amino-3-oxo-propyl] sulfanyl}-L-malate model compound used in the parameterization of the putative hemithioacetal (L)-malate semialdehyde Ec-ASD E241Q covalent reaction intermediate were obtained by RESP fitting[75] to the quantum chemical electrostatic potential of geometry-optimized structures. Hartree-Fock quantum chemical calculations were carried out with GAMESS[76] using the 6–31G* basis set.

Position-dependent amino-acid residue variation in multiple sequence alignment data was analysed using a normalized measure of the Shannon information entropy (H$_X$). The relative Shannon entropy is calculated at each residue alignment position (X) as

$$H_X = \frac{\sum_{i=1}^{20} p_{(i|X)} \ln p_{(i|X)}}{\sum_{i=1}^{20} p_{(i)} \ln p_{(i)}},$$

where $p_{(i/X)}$ is the conditional probability of residue type $(i)$ occurrence at alignment position $(X)$, and $p_{(i)}$ is the normalized probability of residue type $(i)$ occurrence at any position. To minimize sampling bias, $p_{(i)}$ was taken as the globally normalized residue type probability values for all natural proteins tabulated by Ranganathan and co-workers[77].

**Data availability.** The authors declare that all the data supporting the findings of this study are available within the paper and its Supplementary Information Files, as well as from the authors upon reasonable request.

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

## Acknowledgements

This work was supported by the French National Research Agency (ANR programme d'Investissement d'Avenir, Project SYNTHACS, ANR-10-BTBR-05-01) and by Adisseo. We are grateful to S. Bozonnet and S. Pizzut-Serin for providing support in using the ICEO high-throughput facility at the LISBP, devoted to the engineering and screening of new and original enzymes, and to A. Vax, H. Serrano-Bataille, F. Calvayrac, and J. Fredonnet for their technical assistance. This work was granted access to the HPC resources of the Computing Center of Region Midi-Pyrénées (CALMIP, Toulouse, France).

## Author contributions

T.W. and J.M.F. originally proposed the novel metabolic pathway for DHB production. T.W., M.M., R.H., I.A., M.R.-S., and J.M.F. conceived the general structure of the project. T.W., R.I. and C.A. carried out the enzyme and strain engineering. A.B., H.C., C.D., L.L.-H., N.M., M.N. and Y.M. contributed to the analysis of the enzymes and engineered strains. C.M.T., N.T., R.I., M.R.S. and I.A. performed the enzyme structural analyses, molecular modelling, design of enzyme libraries and screening assays. T.W., C.M.T., R.I., I.A., M.R.-S. and J.M.F. wrote the manuscript. All authors read and approved the manuscript.

## Additional information

**Competing interests:** M.M. and R.H. are employees of the ADISSEO Company that provides financial support to this study. T.W., C.D., H.C. and J.M.F. are coauthors on a patent based upon this work WO2013/160762A2 (2013). In addition, T.W., C.M.T., M.M., R.H., I.A., M.R.-S. & J.M.F. are coauthors of a patent that described the present work WO2012/056318A1, that has been granted January 19th, 2016 as US 9,238,829 B2. All remaining authors declare no competing financial interests.

