## [Peer Review File · Nature Communications]

REVIEWERS' COMMENTS:

Reviewer #1 (Remarks to the Author):

Authors developed a three-step metabolic pathway for the synthesis of 2,4-Dihydroxybutyric acid (DHB) starting from the natural metabolite malate. The pathway employs previously unreported malate kinase, malate semialdehyde dehydrogenase, and malate semialdehyde reductase activities by rational design based on structural and mechanistic knowledge of candidate enzymes acting on sterically cognate substrates. The pathway was expressed in a minimally engineered *Escherichia coli* strain and produced 1.8 g/l DHB with a molar yield of 0.15.

Since the production of methionine via DHB increases the theoretical yield by approximately 100 % compared to the conventional one-step biosynthesis of methionine from glucose and sulphate, and by approximately 30 % when compared to the biosynthesis of methionine from glucose and methanethiol. This study is very interesting and should be published after following issues are addressed:

1. Beside the AK, encoded by *lysC18*, and ASD, encoded by *asd19*, from *E. coli*, and HSD, encoded by *HOM6*, from *Saccharomyces cerevisiae* since the HSD enzymes in *E. coli* are bifunctional enzymes with an associated AK activity, are there any other enzymes that could have the required catalytic activities for DHB? Why only these few enzymes were selected?

2. To test this hypothesis, saturation mutagenesis of *Ec-Asd* was carried out at residue position 241. The reason of selecting position 241 needs further clarification.

3. Page 13, lines 361-362: I do not see enough evidences that DHB biosynthesis is rate limited by MSD activities? Other enzymes should also be important.

Reviewer #2 (Remarks to the Author):

The paper concerns the re-engineering of a biochemical pathway to allow the production of a synthetic metabolite, 2,4-dihydroxybutanoic acid, in *E. coli* through the application of synthetic biology approaches. This research represents a significant achievement, advancing the field considerably, and demonstrating the power of metabolic engineering for the production of commodity and fine chemicals. This is the kind of research that should make its way into text-books. The authors have followed a logical and well-thought out strategy to achieve their goals and in so doing have undertaken a huge amount of work. In essence, they have manipulated the pathway for homoserine production to allow it to be used for dihydroxybutyrate biosynthesis. It mirrors the pioneering work of John Holbrook from the 1980's in the redesign of an enzyme activity (lactate dehydrogenase) – but here the authors have used a similar strategy to redesign three enzymes of a pathway to accept new substrates and allow the metabolism to take place with very good yields. This represents a major breakthrough in the field. I have only a few minor comments on a truly exceptional paper:

There are some comments the authors may wish to consider:

1. With respect to *Ec-LysC*, the best single mutation would appear to be E119A, yet the E119S variant is incorporated into the triple mutant? Any particular reason this?

2. I'm not sure that Fig 2B tells me anything – I'd prefer to see figures more like shown with Fig 3A. In fact, there is room to be a bit more consistent with how the data is presented.

3. On page 8, it is stated that malyl-P is unstable – does this instability interfere with the yield of the final product – ie is there room to improve the yield of the final product by ensuring the malyl-P is quickly utilized by the next enzyme?

4. In places it is not clear why some mutations offer advantages – eg page 9, last paragraph: why does conversion of glutamate residues to glutamines improve specificity? Is there a chemical logic to this?

5. In the final discussion, there is room for a bit more perspective here of the significance of the work – eg relating to how enzyme specificity changes can be applied to make altogether unnatural metabolites. This would allow referencing of the Holbrook work (Science. 1988 Dec 16;242(4885):1541-4).

REVIEWERS' COMMENTS:

Reviewer #1 (Remarks to the Author):

Authors developed a three-step metabolic pathway for the synthesis of 2,4-Dihydroxybutyric acid (DHB) starting from the natural metabolite malate. The pathway employs previously unreported malate kinase, malate semialdehyde dehydrogenase, and malate semialdehyde reductase activities by rational design based on structural and mechanistic knowledge of candidate enzymes acting on sterically cognate substrates. The pathway was expressed in a minimally engineered *Escherichia coli* strain and produced 1.8 g/l DHB with a molar yield of 0.15. Since the production of methionine via DHB increases the theoretical yield by approximately 100 % compared to the conventional one-step biosynthesis of methionine from glucose and sulphate, and by approximately 30 % when compared to the biosynthesis of methionine from glucose and methanethiol. This study is very interesting and should be published after following issues are addressed:

1. Beside the AK, encoded by *lysC18*, and ASD, encoded by *asd19*, from *E. coli*, and HSD, encoded by *HOM6*, from *Saccharomyces cerevisiae* since the HSD enzymes in *E. coli* are bifunctional enzymes with an associated AK activity, are there any other enzymes that could have the required catalytic activities for DHB? Why only these few enzymes were selected?

We are grateful to the Reviewer for their constructive and positive comments, and we reply below to each point.

Regarding the third reaction step, catalyzed by malate semialdehyde reductase, we have actually screened nine different aldehyde reductases (including *Hom6*) for the required activity (see supplementary Table 7). Significant MSA reductase activity was identified in three enzymes - one of which was further improved by rational engineering. In this case, we were able to exploit the well-known broad substrate specificity of this class of enzymes that are involved in the intracellular detoxification of a wide variety of aldehydes, and to take advantage of the existence of the comparatively large number of different aldehyde reductases that reduce terminal aldehyde groups on substrates sterically cognate to malate semialdehyde.

We therefore consider it quite likely that other enzymes (not present in our screen) would possess basal malate semialdehyde reductase activity.

In contrast, it is most unlikely in our opinion that malate kinase or phosphorylating malate semialdehyde dehydrogenase enzyme activities could have been identified by extended the screening regime to other candidate cognate enzymes (aside from enzymes presenting potentially useful alternative molecular templates from which to engineer the new activity). Our reasoning is that the characteristic electrostatic properties of the substrate in conjunction with the intrinsically chemically more complex nature of the required reactional transformations in steps 1 and 2 will place severe limitations on the number of potential candidate enzymes. For instance, we observed that kinases that phosphorylate carboxyl groups are quite rare in metabolism. Furthermore, the second (non-phosphorylated) carboxyl group of malate carries negative charge, which would be expected to require neutralisation through complementary electrostatic interactions with the enzyme (commonly mediated by the guanidinium group of an arginine residue in the active site) in

order to confer correct binding of the substrate and therefore activity of the enzyme. Enzymes that catalyze phosphorylation of dicarboxylic substrates occur even less frequently in nature. Apart from the aspartate kinase enzymes that we have engineered in our work, we are only aware of glutamate kinases that fulfill this requirement. However, since glutamate has the same charged chemical groups as aspartate, and given the presence of similar active site configurations in the glutamate and aspartate kinases, we had no reason to believe that this class of enzymes would afford any advantage. Indeed taking account of the larger size of glutamate compared to aspartate or malate, the engineering of malate kinase activity is expected to be more difficult than in glutamate kinases.

Likewise, glutamylsemialdehyde dehydrogenase is the phosphorylating semialdehyde dehydrogenase whose substrate most closely resembles malate semialdehyde glutamylsemialdehyde dehydrogenase. Again, the active site configuration of this enzyme is very similar to the engineered Asd, and we had no reason to believe that it would be more promising to screen and engineer this class of enzymes.

2. To test this hypothesis, saturation mutagenesis of Ec-Asd was carried out at residue position 241. The reason of selecting position 241 needs further clarification.

We are unsure as to why the referee requests further explanation for the selection of position Glu241 in Ec-Asd as a target for mutagenesis. We feel that the preceding paragraph of the manuscript (Page 8, lines 5-17, reproduced below) provides a thorough explanation for this choice.

*“The binding of the ASA α -amino and α -carboxylate groups in the Ec-Asd active-site occurs via salt-bridge interactions with oppositely charged Glu241 and Arg267 residue side-chains. This salt bridging arrangement is similar to that of the (L)-aspartate substrate α -amino and α -carboxylate groups in the complex with *E. coli* aspartate kinase III²³ that catalyses the preceding reaction step in the physiological pathway. The 2-OH group in a malyl-P/MSA substrate/product couple might be able to hydrogen bond with Glu241 in Ec-Asd thereby providing for substrate binding similar to that of the natural substrate derivative in the experimental complex. However, the poor observed activity of wild-type Ec-Asd on MSA compared to ASA may be in part due to a lowering in the binding affinity for an alternative substrate carrying net negative charge. Replacement of the conserved Glu241 residue in the wild-type *E. coli* enzyme by residues with uncharged side-chains would then be expected to improve MSA binding affinity and reduce that of ASA.*

To test this hypothesis, saturation mutagenesis of Ec-Asd was carried out at residue position 241.”

3. Page 13, lines 361-362: I do not see enough evidences that DHB biosynthesis is rate limited by MSD activities? Other enzymes should also be important.

We agree with the Reviewer that the corresponding sentence is unclear. We have therefore changed it to read as written below (now line 9 to 12 in page 14):

“Taken together our results show that overexpression of anaplerotic Ppc activity is a key factor in enabling cellular DHB production, and that increasing basal levels of MSD activity is a major requirement for further improving DHB production.”

Reviewer #2 (Remarks to the Author):

The paper concerns the re-engineering of a biochemical pathway to allow the production of a synthetic metabolite, 2,4-dihydroxybutanoic acid, in *E. coli* through the application of synthetic biology approaches. This research represents a significant achievement, advancing the field considerably, and demonstrating the power of metabolic engineering for the production of commodity and fine chemicals. This is the kind of research that should make its way into textbooks. The authors have followed a logical and well-thought out strategy to achieve their goals and in so doing have undertaken a huge amount of work. In essence, they have manipulated the pathway for homoserine production to allow it to be used for dihydroxybutyrate biosynthesis. It mirrors the pioneering work of John Holbrook from the 1980's in the redesign of an enzyme activity (lactate dehydrogenase) – but here the authors have used a similar strategy to redesign three enzymes of a pathway to accept new substrates and allow the metabolism to take place with very good yields. This represents a major breakthrough in the field. I have only a few minor comments on a truly exceptional paper:

There are some comments the authors may wish to consider:

- 1. With respect to Ec-LysC, the best single mutation would appear to be E119A, yet the E119S variant is incorporated into the triple mutant? Any particular reason this?**

We wish first of all to thank the reviewer for the constructive remarks and warm recommendation to publish our work in Nature Communications.

Reviewer 2 correctly notes that the E119A point mutant has a slightly improved k_{cat}/K_m value compared to E119S. However, in order to reduce the size of the library that could be screened within a reasonable time period, in addition to glycine, we restricted the choice of residue type at position 119 to those with sidechains (Ser, Asn, Asp, Gln) with functional groups potentially able to engage in a hydrogen bond interaction with the malate substrate -2OH hydroxyl (see supplementary Table 6). As the final triple mutant (Lysc V115A E119S E434V) was considered to be sufficiently efficient for pathway implementation, we chose not to further investigate the effect of E119A in combination with V115A and E434V

- 2. I'm not sure that Fig 2B tells me anything – I'd prefer to see figures more like shown with Fig 3A. In fact, there is room to be a bit more consistent with how the data is presented.**

We would like to keep FIG2B in the main manuscript because this figure highlights in an effective visual way the striking loss of all activity towards (L)-aspartate in the V115A:E119S:E343V triple mutant and the reutilization of the substrate binding energy to yield an activity towards (L)-malate (expressed as k_{cat}/K_m) in the engineered enzyme of almost half of that of the cognate enzyme towards the natural substrate, and we prefer to report detailed kinetic data for mono-mutant variants mainly on (L)-malate alone in Supplementary Table 6, where they can be readily interpreted by the reader. Furthermore, we wish to present the data in a way which is to our point of view the best for each one, as the magnitude of observed changes and the available information are very different from each to other.

3. On page 8, it is stated that malyI-P is unstable – does this instability interfere with the yield of the final product – ie is there room to improve the yield of the final product by ensuring the malyI-P is quickly utilized by the next enzyme?

The referee asks a very interesting question. Indeed, excessive degradation of pathway intermediates (and in particular malyIP) may reduce the yield of the synthetic pathway. However, thermodynamic analysis tells us that the equilibrium of the malate kinase reaction is far on the side of the substrates (See supplementaryNote1 and supplementary Table 1). Therefore, intracellular malyIP concentrations are likely to remain low (in the μM range at equilibrium when considering Gibbs standard free energy for the reaction and realistic estimations of the malate, ATP and ADP concentrations) thereby strongly reducing potential losses due to spontaneous degradation of this compound.

In places it is not clear why some mutations offer advantages – eg page 9, last paragraph: why does conversion of glutamate residues to glutamines improve specificity? Is there a chemical logic to this?

The present manuscript focuses on the development of the synthetic pathway rather than on the engineering of three individual enzymes *per se*. For this reason the screening methods and mechanistic interpretations of the effects of the introduced mutations on individual enzymes have been described as concisely as possible so as to avoid deflection of the primary focus of the paper away from the full pathway.

More specifically, a companion paper in which we provide a detailed description of the screening method and a mechanistic discussion of the effect of favorable mutations on aspartate kinase encoded by *lysC*, is in preparation for submission elsewhere.

The Reviewer raises a specific question concerning the reason for the positive effect of the Glu to Gln mutation in *Asd* in respect to the specificity of the mutant enzyme for malyIP. This effect is first observed for *Ec-Asd* (page 8 line 18 to page 9, line 6) and then reproduced for the *BsAsd* and *MjAsd* enzymes (page 9 last paragraph).

We argue that the electrostatic interaction between the negatively charged glutamate 241 and the positively charged amino group in aspartylP facilitates effective binding of the wild-type enzyme with its natural substrate (see on page 8, lines 5-17). Upon mutation of Glu241 to uncharged Gln the strength of this favourable electrostatic interaction with the amino group in aspartylP will be markedly diminished, leading to a decreased activity of the mutant enzyme on the natural substrate.

To better explain this to the reader, we now include a more explicit discussion of the observed effects in the first paragraph on Page 9 of the MS in which the effects of Glu241Gln/Cys mutations in *Ec-asd* are described (marked in red). We make further reference to the *Ec-Asd* Glu241Gln/Cys mutants in the first paragraph on Page 10 where the *Bs Asd* Glu218Gln/Cys and *MjAsd* Glu210Gln/Cys mutations are discussed.

Page 9, first paragraph:

“We found that the introduction of mutations E241Q and E241C respectively improved enzyme specificity by 71- and 17-fold in favour of the malyI-P substrate (Figure 3A). However, the change in specificity was brought about by a marked decrease in activity towards the natural substrate

aspartyl-P, rather than an increase in the intrinsic activity towards malyl-P (Figure 3B, Supplementary Table S6.1). This result showed on one hand that the electrostatic interaction between the negatively charged Glu241 and the positively charged amino group of the natural aspartyl-P substrate was a major requirement for the wild-type activity of this enzyme. On the other hand, it became clear that MSD activity in Ec-Asd could not be significantly increased through the simple restoration of complementary polar interactions between the α -hydroxyl group of malyl-P and alternative amino acid residue types in position 241.”

Page 10, first paragraph:

“However, by analogy to the effects observed in Ec-Asd the reaction specificity of these enzymes is significantly improved in favour of malyl-P when the conserved active-site glutamate residues (Glu218 in Bs-Asd, Glu210 in Mj-Asd) are mutated to glutamine or cysteine.”

- 4. In the final discussion, there is room for a bit more perspective here of the significance of the work – eg relating to how enzyme specificity changes can be applied to make altogether unnatural metabolites. This would allow referencing of the Holbrook work (Science. 1988 Dec 16;242(4885):1541-4).**

We have extended the discussion by emphasizing that our study takes the development of synthetic enzymatic pathways from a conceptual stage to a level where the experimental proof of principle has been demonstrated.

We cite the pioneering work of John Holbrook and colleagues who first showed that rational engineering could be used to generate new enzyme functions.

Please find the modifications to the Discussion (first two paragraphs on page 14).